# LARM: Large Auto-Regressive Model for Long-Horizon Embodied Intelligence

**Zhuoling Li** [1]  **Xiaogang Xu** [2]  **Zhenhua Xu** [3]  **Ser-Nam Lim** [4]  **Hengshuang Zhao** [1]

https://lizhuoling.github.io/LARM_webpage/

## Abstract

Recent embodied agents are primarily built based on reinforcement learning (RL) or large language models (LLMs). Among them, RL agents are efficient for deployment but only perform very few tasks. By contrast, giant LLM agents (often more than 1000B parameters) present strong generalization while demanding enormous computing resources. In this work, we combine their advantages while avoiding the drawbacks by conducting the proposed referee RL on our developed large auto-regressive model (LARM). Specifically, LARM is built upon a lightweight LLM (fewer than 5B parameters) and directly outputs the next action to execute rather than text. We mathematically reveal that classic RL feedbacks vanish in long-horizon embodied exploration and introduce a giant LLM based referee to handle this reward vanishment during training LARM. In this way, LARM learns to complete diverse open-world tasks without human intervention. Especially, LARM successfully harvests enchanted diamond equipment in Minecraft, which demands significantly longer decision-making chains than the highest achievements of prior best methods.

## 1. Introduction

In recent years, remarkable progress has been achieved in various artificial intelligence (AI) topics (LeCun et al., 2015) like computer vision (He et al., 2016) and natural language processing (Kenton & Toutanova, 2019), but most of them lack the capacity to physically interact with the real world. To address this disconnect, the concept of embodied AI is introduced (Chrisley, 2003). Early embodied agents are predominantly developed on simulation platforms for specific tasks such as object grasping and indoor navigation (Savva et al., 2019). While notable advancements are achieved, these agents tend to be specialist models confined to isolated tasks (Huang et al., 2023). To overcome this limitation, recent studies, including this work, employ Minecraft (Baker et al., 2022; Fan et al., 2022; Guss et al., 2019; Wang et al., 2024; Li et al., 2023) as a benchmark to explore embodied agents with open-ended objectives and long-horizon reasoning chains.

The early methods for developing such agents primarily rely on reinforcement learning (RL) (Fan et al., 2022). Due to the limited exploration efficiency of RL, these methods require careful reward engineering for different tasks, and the derived RL policies can mostly only complete a single simple task (Yuan et al., 2023). The advantage of RL policies is that they are usually lightweight for real-time deployment. Differently, recent embodied works begin to investigate large language models (LLMs) (Brown et al., 2020). Owing to the extensive general knowledge and formidable reasoning capabilities of LLMs, these methods demonstrate promising results with significantly reduced domain-specific engineering efforts (Wang et al., 2023a). Nevertheless, LLMs continue to exhibit several limitations. First of all, the outputs of LLMs are usually sentences or code (Zhao et al., 2024) generated through iterative token prediction, necessitating $N$ inference operations for $N$ tokens. Therefore, the response speeds of LLMs are restricted. Secondly, recent research suggests that a huge model size is important for an LLM to generalize well (Achiam et al., 2023), while the computing resource for embodied agents is usually very limited. Our analysis reveals that while giant LLMs with more than 1000B parameters like GPT-4 (Achiam et al., 2023) can answer questions about exploration and crafting issues in Minecraft well, the performance of lightweight LLMs such as LLaVA-7B (Liu et al., 2024a) is limited.

As illustrated in Fig. 1, we aim to combine the advantages of both RL methods and LLM methods while avoiding their drawbacks. To this end, we first propose **L**arge **A**uto-**R**egressive **M**odel (LARM), the main body of which shares the same structure as lightweight LLMs like TinyLLaVA (Zhou et al., 2024). This choice enables us to first pre-train it utilizing numerous webpage data to provide it with basic general knowledge. Taking environmental observation

---

[1]The University of Hong Kong [2]The Chinese University of Hong Kong [3]Tsinghua University [4]University of Central Florida. Correspondence to: Hengshuang Zhao <hszhao@cs.hku.hk>.

*Proceedings of the 42^st International Conference on Machine Learning*, Vancouver, Canada. PMLR 267, 2025. Copyright 2025 by the author(s).

as input, LARM predicts the next action to perform in an auto-regressive manner. Instead of generating a descriptive sentence composed of multiple tokens, LARM directly produces a single token to select the next action, which makes LARM respond more swiftly than common LLMs.

The following problem is how to train LARM. We find that classic RL algorithms cannot train LARM effectively and mathematically reveal this is because the reward feedback gradually vanishes in long-horizon embodied exploration. This phenomenon can be empirically understood as even though a policy selects the correct action, it obtains positive feedback only after the target task is completed, meaning many iterations of delay. In addition, any wrong decision-making in future iterations will cause the policy to get no positive reward, which hides the value of the current correct action. To handle this problem, we introduce referee RL. Its core idea is that we employ a referee (like a giant LLM) to provide immediate feedback about whether the just performed action brings positive contribution to realizing the final target. In this way, we efficiently distill the concerned generalizable knowledge of giant LLMs into our lightweight end-to-end LARM policy during online exploration without human supervision. This marks the first attempt that optimizes an LLM-style embodied policy through making it directly interact with the environment online.

We validate our method in both MineDojo (Fan et al., 2022) and Mineflayer (PrismarineJS., 2013) environments. The experimental results suggest that our method completes diverse challenging tasks with a single model, indicating its promising generalization. LARM achieves higher success rates than previous counterparts, although these counterparts may employ a special network for each task. Notably, LARM is the first method that harvests enchanted diamond equipment in Minecraft. In addition, evaluated with an RTX4090 GPU, LARM runs with a speed of 0.58 second per inference, which meets the speed requirement of online high-level action scheduling.

## 2. Related Work

**Minecraft agents.** Compared with other embodied benchmarks, Minecraft is an open-ended platform suitable for exploring building agents with long-horizon planning capabilities (Fan et al., 2022). It simulates diverse weather, biomes, and mobs in an unlimited 3D virtual world. Early methods in Minecraft are mostly based on reinforcement learning (Frazier & Riedl, 2019) or imitation learning (Baker et al., 2022). Their model outputs are atom actions, *e.g.*, a short movement, mouse click, or keyboard press. However, due to the huge decision space, such atom-based agents are quite challenging for optimization. Thus, these works pay their main attention to devising strategies for alleviating the optimization complexity (Scheller et al., 2020). An effective

practice is devising the policy into a hierarchical architecture, where a complex task is first decomposed into many simple sub-tasks. Different models are trained for various sub-tasks and a leader model is built to decide the order of performing these sub-tasks (Liu et al., 2024b).

Due to its open-world characteristic, Minecraft is suitable for exploring how to develop open-ended embodied intelligence (Feng et al., 2024). To concentrate on studying this problem, there are plentiful works that take a skill (*e.g.*, chopping down a tree or crafting a table) as the basic model output (Wang et al., 2023a). The skill could be modeled as a well-trained policy based on reinforcement learning (Yuan et al., 2023), an action generation model based on language prompt (Lifshitz et al., 2023), or provided APIs. Among these works, LLM-based methods achieve the most impressive results thanks to their rich general LLM knowledge (Achiam et al., 2023; Wang et al., 2023b), especially for giant LLMs with more than 1000B parameters like GPT-4. However, due to the huge model sizes, these LLMs can only be deployed in remote computing clusters. There are also works that try tuning a lightweight LLM like LLaMA using Minecraft relevant text and then prompting the tuned LLM to say what skill should be performed in inference (Feng et al., 2024). However, the text used for tuning contains much irrelevant information and is not specialized for task execution. As the embodied text tuning data volume is also limited, the tuned LLMs often fail to describe what skill should be performed correctly.

**Large language models.** LLMs draw broad attention from the research and industrial communities due to their rich general knowledge and the ability to generate the answers to diverse kinds of questions (Chang et al., 2024). GPT-3 emerges as a milestone in the evolution of LLMs, as it takes the next token prediction problem as the pre-training task and showcases remarkable open-world generalization capabilities (Brown et al., 2020). Subsequently, the fine-tuning of GPT-3 using reinforcement learning with human feedback leads to the creation of ChatGPT (OpenAI, 2023) and GPT-4 (Achiam et al., 2023). However, a significant limitation of LLMs is their inability to interpret information in images, which are vital for humans to perceive the world. To overcome this problem, researchers devise strategies that inject vision information into LLMs and enable LLMs to perceive images. A common method is fine-tuning a small number of network parameters using numerous language-image data pairs to bridge the representation gap between text and images (Ding et al., 2023). In this way, some large vision-language models like LLaVA (Liu et al., 2024a) and Flamingo (Alayrac et al., 2022) are derived.

**RL with LLMs in embodied AI.** RL can search promising decision-making policies without human intervention, and LLMs are able to provide suitable search start points

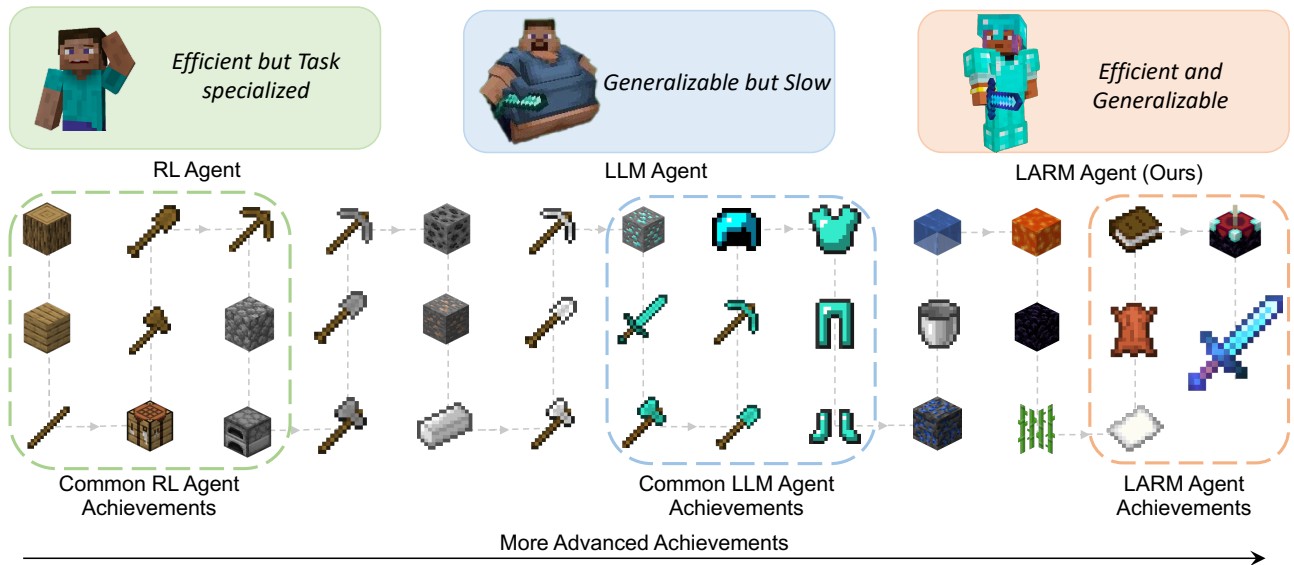

*Figure 1.* Comparison among agents based on RL, LLM, and LARM. As shown, RL agents are usually task specialized, and LLM agents are computationally expensive to deploy. By contrast, the LARM agent is efficient and generalizable. Besides, LARM presents better performance. As shown, LARM is the first method that achieves enchanted diamond equipment in Minecraft.

based on their rich general knowledge (Laleh & Ahmadabadi, 2024). Therefore, it is natural to design ways to combine them to build advanced embodied intelligence. In previous works, a popular choice is training a network for each basic skill based on RL, and then prompting the LLM to say what skill should be used according to the task target and environment observation (Wang et al., 2023b). Nevertheless, this paradigm is not only slow, it requires the LLM to have sufficient knowledge about Minecraft. According to our analysis, giant LLMs own such an ability but the capabilities of lightweight LLMs are limited. Another possible choice is utilizing LLM to generate the code for calculating RL reward (Xie et al., 2024). Nevertheless, it is not always feasible to define a reward function by writing code. For example, in Minecraft, the information is represented as image and agent status information, which cannot be mapped as reward based on rules. To handle this problem, we propose to directly employ GPT-4 to read the agent status before and after executing a skill and judge whether the outcome brought by this skill contributes to realizing the given target. In addition, to the best of our knowledge, this is the first work that directly optimizes an LLM-style policy based on online exploration and reinforcement learning. Our results suggest that the rich general knowledge in LLM favors this exploration and self-learning process.

## 3. Preliminary

### 3.1. Problem Formulation

What we study in this work can be conceptualized as an auto-regressive prediction problem involving long sequences, and is effectively framed as a Markov Decision Process symbolized by a tuple $\mathcal{E} = (\mathcal{S}, \mathcal{A}, \mathcal{P}, \mathcal{T}, \mathcal{R}, \gamma, \tau)$. Specifically, $\mathcal{S}$ is the set of all potential states. $\mathcal{A}$ is the action set, and every action is also called as a skill in this work. $\mathcal{P} : \mathcal{S} \times \mathcal{A} \times \mathcal{S} \to [0, 1]$ represents a probability distribution that governs the state transitions given states and actions. $\mathcal{T}$ is the set of all task targets. $\mathcal{R} : S \to \mathbb{R}$, $\gamma$, and $\tau$ denote the reward function, discount factor, and initial state distribution, respectively. At any discrete time step $t$, the environment resides in a state $s_t \in \mathcal{S}$, and the corresponding observation $o_t$ by a policy $\pi$ is a function of this state, expressed as $o_t = f(s_t)$. This observation $o_t$ is then utilized to select the subsequent action according to $a_t \sim \pi(o_t, \iota)$, where $a_t \in \mathcal{A}$ and $\iota \in \mathcal{T}$ denotes the given target task.

In tackling the studied long-horizon embodied task, the objective is to navigate through a sequence of intermediate states $\tau, s_1, s_2, \ldots, s_{T-1}$ to ultimately reach the target state $s_T$ at the final time step $T$. This requires the policy to generate a series of actions $a_0, a_1, \ldots, a_{T-1}$ such that each action $a_t$ transitions the environment from state $s_t$ to the next state $s_{t+1}$ correctly, adhering to the dynamics prescribed by the transition probability distribution $\mathcal{P}$. It is crucial that each intermediate state $s_t$ is accurately achieved in sequence to ensure the policy attains the target state $s_T$.

### 3.2. PPO

Proximal Policy Optimization (PPO) (Schulman et al., 2017) is a model-free reinforcement learning algorithm widely adopted for training policies in complex environments, and our referee RL is developed based on this algorithm. A PPO

policy $\pi$ mainly consists of two components, the actor $\pi_a$ and critic $\pi_c$. $\pi_c$ is to estimate the value function $V_{\theta_c}(s_t)$, the expected cumulative discounted reward starting from state $s_t$ and following $\pi$. The optimization objective of $\pi_c$ is as follows:

$$L_{\theta_c}^c = \mathbb{E}_t[(V_{\theta_c}(s_t) - (r_t + \gamma V_{\theta_c}(s_{t+1})))^2], \quad (1)$$

where $r_t \in \mathcal{R}$ is the reward received after taking action $a_t$ in state $s_t$. To further reduce the variance of the value estimate and improve stability, PPO employs the generalized advantage estimation (GAE), which is defined as:

$$A_t = \sum_{k=t}^{T-1} (\gamma\lambda)^{k-t}\delta_k, \quad (2)$$

where $\lambda$ is a factor balancing between temporal difference (TD) learning and Monte Carlo estimation, and $\delta_k$ denotes the TD-error, which is formulated as $\delta_k = r_k + \gamma V_{\theta_c}(s_{k+1}) - V_{\theta_c}(s_k)$. With $A_t$, the objective function for the critic $\pi_a$ can be given as:

$$L_{\theta_a}^a = \mathbb{E}_t[\frac{\pi_a(a_t|s_t)}{\pi_a^{old}(a_t|s_t)} A_t], \quad (3)$$

where and $\pi_a^{old}$ is the old actor policy before weight update and $\pi_a(a_t|s_t)$ is the current actor. To improve the training stability, PPO further develops a clipped surrogate objective based on Eq. (3), which can be formulated as:

$$\tilde{L}_{\theta_a}^a = \mathbb{E}_t[\min(k_t A_t, (k_t, 1 - \epsilon, 1 + \epsilon) A_t)], \quad (4)$$

where $k_t = \frac{\pi_a(a_t|s_t)}{\pi_a^{old}(a_t|s_t)}$ and $\epsilon$ is a small positive parameter. By optimizing $\pi_c$ and $\pi_a$ with respect to Eq. (1) and Eq. (4), the policy gradually learns to perform the target task.

## 4. Method

### 4.1. Referee Reinforcement Learning

In long-horizon embodied task exploration, the policy can usually only get positive feedback after the target task is completed successfully (Fan et al., 2022). Following the notations in Section 3.1, we can assume that there is an exploration trajectory $\{(s_k, a_k, r_k)\}_{k=t}^{T}$, where $T$ is a large integer, and $s_k$, $a_k$, and $r_k$ denote the state, action, and environment reward at the $k$ step, respectively. The agent completes the target task at the final step $T$. Therefore, we can get that:

$$r_k = \begin{cases} -\varepsilon, & if \ k = t, t+1, \cdots, T-1 \\ R - \varepsilon, & if \ k = T \end{cases} \quad (5)$$

where $-\varepsilon$ denotes a small negative constant due to time penalty and $R$ is the positive reward of completing the target

**Algorithm 1** Referee RL
___
**Require:** Target task $\iota$
1: Initialize the actor $\pi_a$, critic $\pi_c$, referee $\pi_r$
2: Initialize policy exploration step $T$, policy update steps $N_\pi$
3: **for** each iteration $iter$ **do**
4:    Initialize data buffer $B \leftarrow \emptyset$
5:    **for** $t = 1$ to $T$ **do**
6:        Get the actor observation $o_t \leftarrow f(s_t)$
7:        Get state $s_{t+1}$ and environment reward $r_t$ by taking $a_t \sim \pi_\theta(o_t, \iota)$
8:        Get auxiliary reward $\widehat{r}_t \leftarrow \pi_r(\iota, s_t, a_t, a_{t+1})$
9:        Add transition $B \leftarrow B \cup \{(s_t, o_t, a_t, s_{t+1}, r_t, \widehat{r}_t)\}$
10:   **end for**
11:   **for** $n = 1$ to $N_\pi$ **do**
12:      Sample a random training data batch $\{(s_t, o_t, a_t, s_{t+1}, r_t, \widehat{r}_t)\}_{j=1}^B \sim B$
13:      Optimize $\pi_a$ and with respect to Eq. (1)$\sim$(4)
14:   **end for**
15: **end for**
___

task. As we train the critic $\pi_c$ using the very imbalanced reward set $\{r_k\}_{k=t}^T$ described in Eq. (5) with respect to the optimization objective in Eq. (1), we can infer that the output of $\pi_c$ gradually converges to $V_{\theta_c}(s_k) - \gamma V_{\theta_c}(s_{k+1}) \approx -\varepsilon$. In this way, the corresponding TD error set $\{\delta_t\}_{t=1}^{T-1}$ is:

$$\delta_k \approx \begin{cases} 0, & if \ k = t, t+1, \cdots, T-2 \\ R, & if \ k = T-1 \end{cases} \quad (6)$$

According to the definition of GAE in Eq. (2), we can observe that the first $T - 1 - t$ items are close to zero as the elements in $\{\sigma_k\}_{k=t}^{T-2}$ are nearly zero. For the last item $(\gamma\lambda)^{T-1-t}\delta_{T-1}$, we have $\lim_{T\to\infty}(\gamma\lambda)^{T-1-t} \to 0$ because $\gamma \in (0, 1)$ and $\lambda \in (0, 1)$. Hence, when the task needs long-horizon action chain execution, the obtained GAE value $A_t$ is close to zero, which suggests that the optimization objective for training $\pi_a$ in Eq. (4) becomes zero. In this way, even though the policy makes the right action decision, it cannot get any positive feedback.

To handle this problem, we introduce referee RL. Specifically, we introduce a referee $\pi_p$ to provide an auxiliary reward feedback to the trained policy $\pi$, and this reward at the step $k$ is represented as $\widehat{r}_k = \pi_p(\iota, s_k, a_k, s_{k+1})$. This means $\pi_p$ takes the target task information $\iota$, initial state $s_k$, selected action $a_k$, and new state $s_{k+1}$ as input and provides feedback based on whether the selected action is correct and outcome caused by this action. In this work, we split the feedback into four categories: (a) The selected action is correct and brings a positive outcome to realizing the target. (b) The selected action is correct but does not bring a positive outcome. (c) The selected action is incorrect but

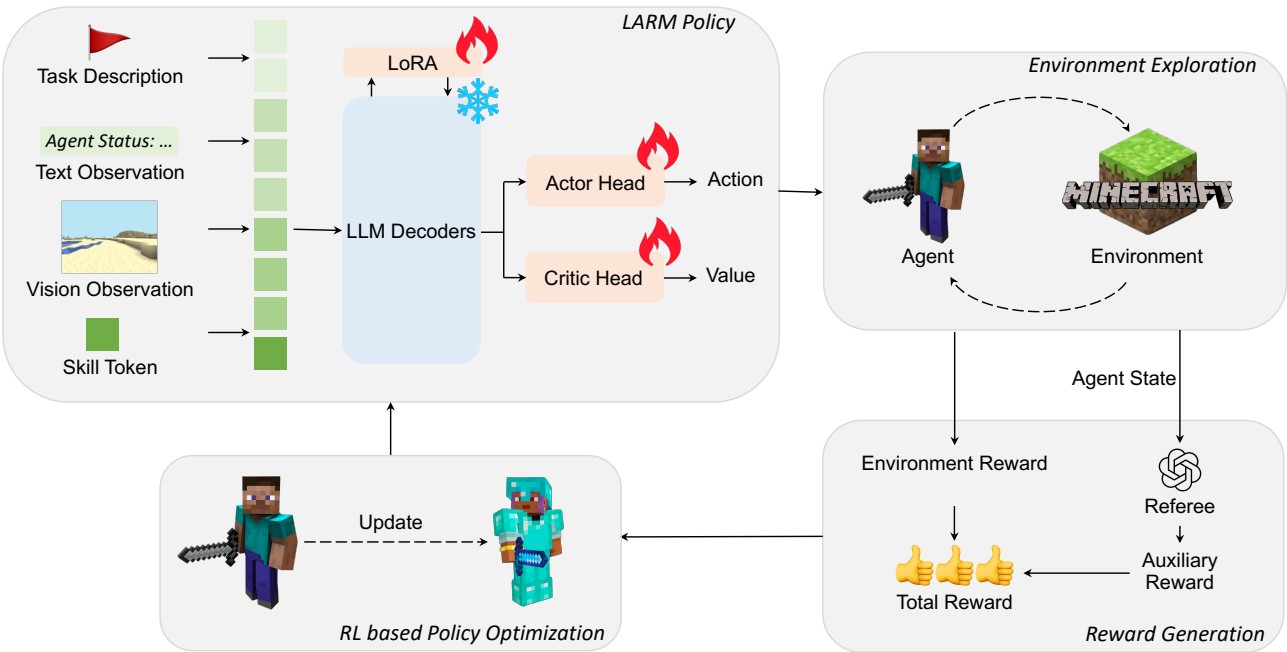

*Figure 2.* The overall pipeline of our method. As illustrated, we parametrize the actor $\pi_a$ and critic $\pi_c$ using a single LARM model with two separate prediction heads, *i.e.*, the action head and critic head. We train LARM based on our proposed referee RL algorithm, which utilizes both environment feedback and referee generated auxiliary reward to guide the optimization of LARM.

does not lead to a negative outcome. (d) The selected action is incorrect and results in a negative outcome. For the four categories, $\pi_p$ correspondingly returns the reward value $r^a$, $r^b$, $r^c$, and $r^d$, where $r^a > r^b > 0 > r^c > r^d$. By adding this auxiliary reward $\hat{r}_k$ to the original reward described in Eq. (5), the feedback to $\pi$ before $T$ does not remain as the constant $-\varepsilon$. In this way, the TD error $\delta_k$ and corresponding GAE value $A_t$ do not converge to zeros, being able to provide effective optimization guidance to $\pi_a$.

In this work, we model the referee $\pi_p$ based on GPT-4 (Achiam et al., 2023), which is a giant LLM and owns extensive generalizable knowledge. As mentioned before, the information provided to GPT-4 includes $\iota$ (target task description), $a_k$ (the executed skill), $s_k$ and $s_{k+1}$ (the inventory list and environment resource surrounding the agent before and after executing $a_k$), and then we prompt it to judge the situation and response a reward among $r^a$, $r^b$, $r^c$, and $r^d$. The full detailed procedure of the referee RL algorithm is elaborated in Algorithm 1.

### 4.2. Large Auto-Regressive Model

In this part, we explain how to parametrize $\pi_a$ and $\pi_c$ using our designed LARM policy. As shown in Fig. 2, the main body of LARM is the decoders of a lightweight decoder-only LLM, TinyLLaVA-3.1B in this work. The parameters of these decoders are frozen during training and a trainable LoRA (Hu et al., 2021) module is applied to help the model learn new knowledge in the applied task domain. This

design has two key benefits: (i) The model is initialized with the general knowledge and reasoning ability of LLMs while maintaining an acceptable parameter volume. In this way, LARM can be deployed based on the restricted computing resources in embodied applications and achieve real-time response. (ii) As LARM adopts a similar model architecture as LLMs, we can first pre-train it using numerous question-answer data related to the concerned embodied AI topics. This kind of data is much easier to collect and scale up than embodied data with action execution. We have tried pre-training LARM using a 34G webpage dataset crawled from Wiki (Fan et al., 2022), and the results indicate that the training convergence is improved.

The input to the LARM model consists of four parts, *i.e.*, task description, text observation, vision observation, and a skill token. The task description specifies the target task to conduct. The text information primarily includes the inventory list, historical action, and blocks surrounding the agent. The vision information is the real-time image perceived by the agent. We encode text and image information as tokens based on CLIP (Radford et al., 2021), and these tokens with an additional learnable skill token are input to the LARM decoders to conduct feature interaction. After the decoders, the skill token is input to the action head and critic head depicted in Fig. 2 to output the action and state value. Therefore, LARM parametrizes the actor $\pi_a$ and critic $\pi_c$ in Section 4.1 based on a single model with two different trainable prediction heads.

Similar to previous literature (Wang et al., 2023a; Liu et al., 2024b), the action predicted by LARM is a skill, such as chopping down a tree or searching for a cobblestone. LARM selects a skill to perform by conducting feature matching between the action head output and skill description like STEVE (Zhao et al., 2024). In this work, we test LARM with two kinds of skills, the RL-based skills in MineDojo and API-based skills based on Mineflayer. The former category is easier to be extended to real-world applications and the latter class presents higher execution success rates. According to the given task requirements, new skills can be dynamically added based on the strategies developed in previous works (Wang et al., 2023a) and how to generate these skills is not the research focus of this work.

Combining the aforementioned techniques, LARM is optimized by iterating between environment exploration and policy update described in Algorithm 1. The training details like the optimizer choice follow PPO (Schulman et al., 2017). For learning to complete the most challenging task in this work (craft an enchanted diamond tool), about 42 hours of exploration is taken using a single RTX4090 GPU.

# 5. Experiments

## 5.1. Environment

In this work, we validate our method using the MineDojo and Mineflayer environments. Both these two environments are developed based on Minecraft, but there exist differences in their testing protocols. In addition, the commonly used strategies of generating skills in these two environments are also different. For fair comparison, we compare our method with the counterparts in these two environments separately using the same testing protocol and skill generation strategy.

**MineDojo.** MineDojo (Fan et al., 2022) is a pioneering benchmark suitable for studying how to develop open-ended and generally capable embodied agents. It features a simulation suite with thousands of tasks specified through natural language prompts. The behaviors that can be conducted in MineDojo primarily include navigation, harvest, combat, and craft. When testing a policy, an agent is randomly initialized in a biome with some initial tools. The policy needs to control the agent to explore and harvest resources in the environment and gradually realize the given target. The action space in MineDojo is a compound multi-discrete space that allows the agent to select a movement action or an optional functional action at each step, encompassing a diverse range of arguments to facilitate complex interactions.

**Mineflayer.** Compared with MineDojo, the basic actions in Mineflayer (PrismarineJS., 2013) are provided APIs, such as harvesting a log or finding the nearest stone. Compared with MineDojo, these APIs help researchers concentrate more on high-level decision making rather than repetitive action details. In this way, several significantly more advanced achievements are obtained by previous works based on Mineflayer, like harvesting diamonds with high success rates (Wang et al., 2023a). In addition, the agents are usually spawned without initial tools in Mineflayer.

## 5.2. Main Results

We compare LARM with previous methods in this part. As the basic actions in MineDojo and Mineflayer are different, we conduct the comparison separately.

**Comparison on MineDojo.** The methods compared on MineDojo include MineAgent (Fan et al., 2022), Plan4MC (Yuan et al., 2023), LLaMA-Rider (Feng et al., 2024), and RL-GPT (Liu et al., 2024b). Among them, MineAgent is the baseline method provided by Minddojo. It first fine-tunes CLIP (Radford et al., 2021) based on numerous web data and uses the fine-tuned CLIP to guide the training of reinforcement learning algorithms. Plan4MC is a reinforcement learning based method. It splits a task into basic skills and trains an agent to learn them one by one in a hierarchical way. LLaMA-Rider is an LLM obtained by fine-tuning LLaMA. It first makes the agent explore the environment to collect data. Then, the collected data is adopted to fine-tune LLaMA in a supervised manner. RL-GPT builds two LLM based agents (a slow agent and a fast agent) to schedule the agent actions. For an action step, this method first queries the agents whether this action step can be completed via code generation. If the code generation is infeasible, an RL based action is performed.

We compare LARM with these methods on diverse tasks, and the detailed settings of these tasks follow previous works (Feng et al., 2024). Notably, we train a single LARM model to complete all the tasks, which is different from many previous works that employ separate models for various tasks. This characteristic suggests the promising generalization ability of LARM. To compute success rates, we test LARM for 30 times on every task. The experimental results are reported in Table 1. As shown, LARM presents higher success rates than the compared counterparts in all the test tasks, and the promising performance of LARM is mainly attributed to our designed referee RL algorithm, which addresses the reward vanishment problem in long-horizon embodied exploration. In addition, we can observe that LARM achieves higher success rates on tasks with shorter action chains, indicating the great challenge in developing long-horizon embodied intelligence. For example, the *Harvest bucket* task requires three iron ingots, and the *Harvest iron sword* demands two iron ingots and one stick. Therefore, *Harvest iron sword* needs one more step (*Harvest stick*) than *Harvest bucket*, which causes that LARM obtains a higher success rate in the task of crafting a bucket.

**Comparison on Mineflayer**. To fully reveal the superiority

*Table 1.* Performance comparison with previous methods based on MineDojo.

| Task | MineAgent | Plan4MC | LLaMA-Rider Base | LLaMA-Rider | RL-GPT | LARM (Ours) |
|---|---|---|---|---|---|---|
| Harvest stick | 0.00 | 0.30 | 0.23 | 0.43 | 0.65 | 0.93 |
| Harvest crafting table | 0.03 | 0.30 | 0.37 | 0.67 | 0.65 | 0.87 |
| Harvest bowl | 0.00 | 0.47 | 0.73 | 0.97 | - | 0.97 |
| Harvest chest | 0.00 | 0.23 | 0.67 | 0.77 | - | 0.83 |
| Harvest wooden pickaxe | 0.00 | 0.03 | 0.00 | 0.37 | 0.67 | 0.70 |
| Harvest wooden sword | 0.00 | 0.47 | 0.63 | 0.10 | - | 0.70 |
| Harvest furnace | 0.00 | 0.37 | 0.00 | 0.17 | 0.67 | 0.73 |
| Harvest stone stairs | 0.00 | 0.47 | 0.00 | 0.57 | - | 0.67 |
| Harvest stone sword | 0.00 | 0.10 | 0.00 | 0.00 | - | 0.40 |
| Harvest iron ingot | 0.00 | 0.47 | 0.03 | 0.13 | - | 0.60 |
| Harvest bucket | 0.00 | 0.20 | 0.00 | 0.00 | | 0.37 |
| Harvest iron sword | 0.00 | 0.20 | 0.00 | 0.00 | - | 0.27 |
| Harvest beef | 0.33 | 0.43 | 0.03 | 0.03 | 0.46 | 0.60 |
| Harvest mutton | 0.35 | 0.33 | 0.00 | 0.03 | 0.38 | 0.63 |

*Table 2.* Performance comparison based on Mineflayer.

| Achievement | AutoGPT | Voyager | STEVE | LARM (Ours) |
|---|---|---|---|---|
| Wooden sword | 3/3 | 3/3 | 3/3 | 30/30 |
| Stone sword | 3/3 | 3/3 | 3/3 | 30/30 |
| Iron sword | 3/3 | 3/3 | 3/3 | 30/30 |
| Diamond sword | 0/3 | 1/3 | 3/3 | 28/30 |
| Enchanted sword | 0/3 | 0/3 | 0/3 | 16/30 |

*Table 3.* Ablation Study on Reward Design.

| Reward | Stick | Wooden | Stone | Iron |
|---|---|---|---|---|
| ER | 0.20 | 0.13 | 0.10 | 0.00 |
| ER+LAR | 0.30 | 0.23 | 0.13 | 0.00 |
| ER+AR2 | 0.80 | 0.53 | 0.20 | 0.07 |
| ER+AR4 | 0.93 | 0.70 | 0.40 | 0.27 |

*Table 4.* Ablation Study on LLM base selection.

| LLM Base | Stick | Wooden | Stone | Iron |
|---|---|---|---|---|
| TinyLLaVA-0.5B | 0.80 | 0.50 | 0.27 | 0.13 |
| TinyLLaVA-3.1B | 0.83 | 0.57 | 0.33 | 0.13 |
| TinyLLaVA-3.1B[*] | 0.93 | 0.70 | 0.40 | 0.27 |

*Table 5.* Analysis on noisy reward.

| Noise Ratio | Stick | Wooden | Stone | Iron |
|---|---|---|---|---|
| 0% | 0.93 | 0.70 | 0.40 | 0.27 |
| 10% | 0.93 | 0.67 | 0.30 | 0.17 |
| 30% | 0.77 | 0.33 | 0.17 | 0.07 |
| 50% | 0.50 | 0.13 | 0.00 | 0.00 |

of LARM, we further evaluate LARM using the Mineflayer based environment. The compared methods include Auto-GPT (autogpt), Voyager (Wang et al., 2023a), and STEVE (Zhao et al., 2024). In this work, Voyager is a training-free method implemented based on GPT-4. Its main contribution is designing a multi-step prompt generation pipeline. When a target task is given, Voyager prompts GPT-4 to know which skill should be executed and gradually realize the target. AutoGPT is an LLM being able to reason which skill should be performed through multi-step question answering. STEVE is a large vision-language model. In STEVE, a dataset including both videos and text-image pairs is gathered and utilized to fine-tune LLaMA (Touvron et al., 2023), and then the fine-tuned model can invoke pre-defined skills.

In this experiment, the agent is spawned in a random biome without initial inventory. The test tasks include *Harvest wooden sword*, *Harvest stone sword*, *Harvest iron sword*, *Harvest diamond sword*, and *Harvest enchanted diamond sword*. As shown in Fig. 1, *Harvest enchanted diamond sword* is significantly more complex than the other achievements. The previous methods usually test their models three times in Mineflayer. To reduce randomness, we run LARM for 30 times. The experimental results are reported

in Table 2. We can observe that LARM outperforms the compared methods in obtaining different levels of achievements. Notably, LARM is the first method that harvests an enchanted diamond sword in Minecraft successfully.

### 5.3. Ablation Study

**Analysis on reward design**. The key difference of Referee RL from the classic PPO implementation is the reward design. Therefore, we ablate different reward settings in this part. We compare four reward choices in MineDojo with four tasks, *i.e.*, *Harvest stick*, *Harvest wooden sword*, *Harvest stone sword*, and *Harvest iron sword*. The four reward choices are ER (environment reward only), ER+LAR (environment reward plus auxiliary reward produced by LLaVA-7B, a lightweight LLM), ER+AR2 (environment reward plus auxiliary reward produced by GPT-4o, but the auxiliary reward is determined based on only whether the action outcome is positive), and ER+AR2 (the standard setting of our method described in this paper). The experimental results are presented in Table 3.

As shown, when only the environment reward is provided, the agent presents a significantly better success rate on *Harvest stick* than *Harvest stone sword*, where the latter task demands a longer action chain. This phenomenon is because that the environment reward gradually vanishes with the in-

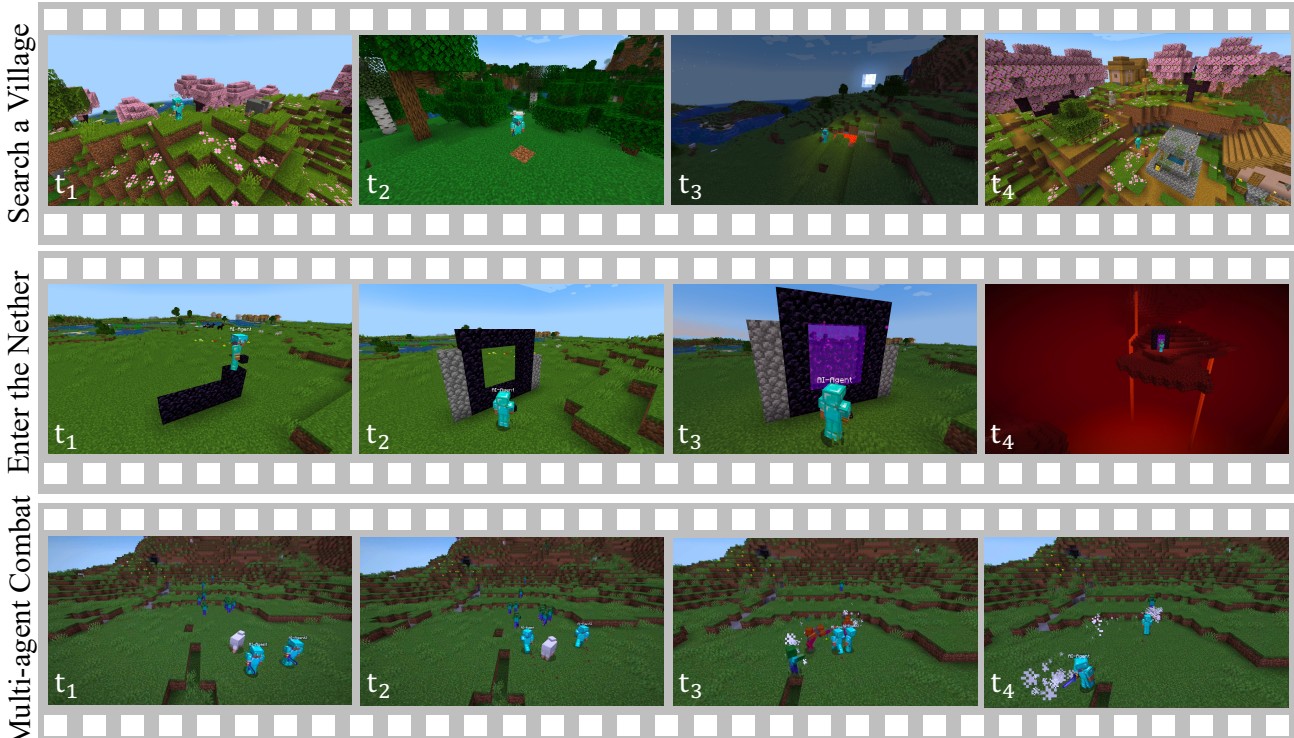

*Figure 3.* More behavior example illustrations of LARM, which include traveling a long distance to find a village, building a nether portal and then entering the nether, multiple agents collaborate with each other to combat zombies.

crease of the action chain length. When the auxiliary reward is generated by LLaVA-7B, the result is also poor, which is because that LLaVA-7B does not understand the Minecraft world well and the provided reward is often incorrect. This result confirms our aforementioned claim that only giant LLMs serve as referees well. In addition, the performance of ER+AR2 is worse than ER+AR4, which is because selecting the correct action does not always bring a positive outcome. For example, the agent decides to search a tree but does not find it successfully. In this situation, ER+AR2 returns a negative reward, which punishes the searching a tree decision even though this choice may be correct. By contrast, ER+AR4 will give a small positive reward.

**Analysis on LLM base selection**. As mentioned before, we utilize the weight of a lightweight LLM to initialize the parameters in LARM to provide initial general knowledge. Then, we fine-tune LARM using Minecraft related webpage data to enhance its understanding. In this experiment, we analyze how the lightweight LLM choice affects the results and whether the webpage data pre-training described in Section 4.2 is beneficial. To this end, we first compare the performances of LARM using TinyLLaVA-0.5B and TinyLLaVA-3.1B as the LLM base, where the experiment results are presented in the 1st and 2nd row of Table 4. As shown, adopting an LLM with more parameters as the LARM model base favors the embodied performance, which is mainly attributed to that a larger LLM has stronger

general knowledge. Then, we compare the performances of models without and with the webpage data pre-training, and the results are reported in the 2nd and 3rd rows of Table 4. It can be observed that webpage data pre-training improves the success rates on embodied task execution. This result suggests that the numerous text and image data on the Internet has the potential to benefit embodied agents.

**Analysis on LLM capability**. We have mentioned that giant LLMs answer questions about the Minecraft world well while the understanding of lightweight LLMs is poor. In this part, we demonstrate this by providing a question-answer example and highlighting the key content in **bold**:

*Prompt: In Minecraft, you need to craft a stone pickaxe. What additional resources do you need to gather if you have only got cobblestones in your inventory?*

*GPT-4o: If you already have cobblestone in your inventory, you need to gather wooden planks to craft sticks. You need **two sticks and three cobblestones** to craft a stone pickaxe.*

*Llama3-8B: You need **a stick**.*

*TinyLLaVA-3.1B: To craft a stone pickaxe in Minecraft, you will need the following additional resources: **cobblestone, stone, wood, leaves, dirt, grass, pillar, shovel, and sword**.*

*TinyLLaVA-3.1B after Webpage data pre-training: You additionally need **two sticks**.*

Comparing the answers, we can observe that webpage data

pre-training significantly improves the concerned domain of knowledge in TinyLLaVA-3.1B.

**Analysis on noisy reward**. In the previous experiments, we show that incorporating a referee into the RL framework improves the exploration efficiency very significantly. However, this is partly because GPT-4 can provide high-quality feedback about whether the executed skill is beneficial to realizing the final target in Minecraft. The effectiveness of referee RL under circumstances that the referee feedback is noisy (many false referee rewards) is not studied. In this experiment, we randomly replace different ratios of referee rewards generated by GPT-4 as other incorrect referee rewards to test the performances of LARM under noisy rewards. The results are presented in Table 5. We can find that when the noise ratio exceeds 10%, the LARM performance drops significantly.

## 5.4. Case Study

The previous experiments mainly show the performances of LARM on harvesting various categories of materials and crafting tools. In Fig. 3, we visualize more examples of other behaviors in the Mineflayer based environment, such as exploring the open world, constructing a building with a specific structure, and multiple agents cooperate to combat dangerous creatures. These behaviors suggest that our proposed techniques in this work have the potential to be further generalized to other diverse domains.

## 6. Conclusion

In this work, we have proposed LARM, which is efficient and possesses general knowledge. To train it, we have revealed the feedback vanishment problem in applying classic RLs to long-horizon embodied exploration. To address this feedback vanishment, we have developed the referee RL technique. By optimizing LARM with referee RL, our method can learn to complete diverse embodied tasks without human supervision. Especially, LARM is the first method that obtains the enchanted diamond equipment achievement in the Minecraft benchmark successfully.

## Impact Statement

This paper presents work whose goal is to advance the field of embodied intelligence. There are many potential societal consequences of our work, none which we feel must be specifically highlighted here.

## Acknowledgments

This work is supported by the National Natural Science Foundation of China (No. 62422606, 62201484).

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
