# OpenReview forum: "LARM: Large Auto-Regressive Model for Long-Horizon Embodied Intelligence"
_ICML.cc/2025/Conference — ICML 2025 poster_

### Official Review · Reviewer_xmFZ · 2025-03-05

**Overall Recommendation:** 2

**Summary:**

In the open-world environment of Minecraft, this paper proposes the Large Auto-Regressive Model (LARM), which leverages the instruction-following and generalization capabilities of large language models to construct a Minecraft agent. Additionally, the paper introduces Referee RL to provide immediate feedback for training LARM. Experimental results on MineDojo and Mineflayer environments show that LARM outperforms baselines.

## update after rebuttal
I have carefully read all the reviewers' comments as well as the authors' rebuttal. Most of my concerns can be addressed through a thorough revision of the manuscript. However, my primary concern remains with the evaluation on the MineRL environment.

While I appreciate the additional experiments provided by the authors, I find the following issues problematic:

1) I am surprised by the authors were able to adapt their method to the MineRL environment, train the LARM model, and complete evaluations on 200 tasks (each with 30 trials) within just one day.

2) Due to the significant lack of implementation details and the experiments on MineRL, the paper requires substantial revision.

So I keep the scores.

**Claims And Evidence:**

One of the key claims of this paper is leveraging the advantages of RL methods and LLMs while mitigating their limitations. Regarding the slow inference speed of LLMs, the paper asserts that LARM achieves an inference speed of 0.58 seconds per inference. However, the paper does not provide a comprehensive experimental comparison for inference speed. Moreover, this speed does not meet the 20Hz inference requirement of MineDojo.

**Essential References Not Discussed:**

The paper does not discuss comparisons between the proposed LARM and key baselines [1] [2] [3] [4] in current Minecraft research. The authors need to clarify the differences or advantages of LARM compared to the aforementioned agents.

[1] Wang, Zihao, et al. "Describe, explain, plan and select: Interactive planning with large language models enables open-world multi-task agents." NeurIPS 2023.

[2] Wang, Zihao, et al. "Jarvis-1: Open-world multi-task agents with memory-augmented multimodal language models." TPAMI 2024.

[3] Qin, Yiran, et al. "Mp5: A multi-modal open-ended embodied system in minecraft via active perception." CVPR 2024.

[4] Li, Zaijing, et al. "Optimus-1: Hybrid multimodal memory empowered agents excel in long-horizon tasks." NeurIPS 2024.

**Experimental Designs Or Analyses:**

As stated in the Methods and Evaluation Criteria section, the paper lacks a sufficient number of evaluation tasks and up-to-date baselines.

**Methods And Evaluation Criteria:**

1. Table 1 presents experiments conducted on MineDojo; however, the number of evaluated tasks is too limited, significantly fewer than the MineDojo benchmark, which includes 1,581 tasks for Programmatic Tasks.

2. There is a lack of comparison to some powerful baselines on MineDojo, such as DEPS [1], Jarvis-1 [2], MP5 [3], etc.

[1] Wang, Zihao, et al. "Describe, explain, plan and select: Interactive planning with large language models enables open-world multi-task agents." NeurIPS 2023.

[2] Wang, Zihao, et al. "Jarvis-1: Open-world multi-task agents with memory-augmented multimodal language models." TPAMI 2024.

[3] Qin, Yiran, et al. "Mp5: A multi-modal open-ended embodied system in minecraft via active perception." CVPR 2024.

**Other Comments Or Suggestions:**

None.

**Other Strengths And Weaknesses:**

Strength

1. Using a large language model to provide rewards is an interesting idea. Although the reward implementation in this paper is relatively simple, it offers new insights into reward design in the field of reinforcement learning.

Weakness

1. As stated in line 218, the paper provides GPT-4 with an inventory list and information about environmental resources surrounding the agent to generate appropriate rewards. At the same time, these information serve as input of LARM. However, this implementation relies on internally integrated environment APIs, making it an unfair comparison against most existing works.

**Questions For Authors:**

1. As stated in line 271, the paper follows Voyager’s skill update strategy but does not explain how these skills are generated. Does this imply that these skills originate from Voyager or that LARM is based on the Voyager framework?

2. Could the authors provide more implementation details on how skills interact with the environment in MineDojo? As far as I know, the action space in MineDojo consists of low-level actions rather than code.

3. How was the 0.65 success rate in Table 1 obtained? Line 340 states that each task was executed 30 times—how does this translate to a success rate of 0.65?

**Relation To Broader Scientific Literature:**

The core objective of this paper is to leverage large language models to generate appropriate skills (code) for interacting with the environment to complete tasks. The fundamental idea is derived from prior work, Voyager. The reviewer did not find significant innovations in terms of model architecture, training methods, or skill implementation. However, the reward design approach may contribute to the community.

**Theoretical Claims:**

No obvious errors were found in the theoretical claims.

---

> ### Author Rebuttal · Authors · 2025-03-31
>
> We believe the Reviewer has significant misunderstandings of this work. In the following, we address the concerns one by one using more precise explanations and sufficient experiments.
>
> ## Q1: Inference speed analysis
>
> We did not explicitly compare the speed of our method with the counterparts because many of them are based on LLMs deployed on remote servers like GPT-4. Their speeds are largely influenced by network latency, so it is difficult to compare fairly.
>
> To conduct a comprehensive speed comparison as suggested, we first report the efficiency metrics of our method as follows:
>
> Inference Time | Inference Memory | FLOPs | Training Time
> | :-: | :-: | :-: | :-: |
> 0.58s | 3.8G | 614.8G | 42 hours
>
> Then, we study how the success rates and inference times are changed by adopting different base LLMs. Due to the reply character limit, please refer to the reply to Q5 of Reviewer 2LZg for results.
>
> For the online inference claim, it is because LARM is for high-level schedule. Each time of high-level schedule corresponds to several seconds of low-level skill execution. The inference speed of the low-level skill policy is more than 1000 FPS. So the LARM speed of 0.58 second per inference meets the online inference requirement. To avoid such misunderstandings, we change the claim in the paper as "meet the speed requirement of online high-level action scheduling".
>
> ## Q2: Evaluated task number
>
> Actually, we have test our method in other tasks of MineDojo. Our method is still very effective.
>
> We do not report results on them because all previous works in MineDojo choose to select representative tasks to report results, although there are totally 1581 tasks in MineDojo. If we report the results on other tasks, we have no method to compare. The reported tasks in Table 1 of the paper follow previous published works.
>
> To further show the effectiveness of our method, we add experiments based on a household simulator VirtualHome and a real-world robot. Refer to the replies to Q1 and Q2 of Reviewer 4ZZV for details.
>
> ## Q3: Missing comparison
>
> As reminded by the Reviewer, we will add all the missing references to the paper and discuss the relation with them.
>
> In the following, we compare success rates with them. As their experiment settings are different from ours, we unify all methods using the setting of DEPS. The success rates on key achievements are as follows:
>
> | Method | Wooden | Stone | Iron | Diamond |
> | :-: | :-: | :-: | :-: | :-: |
> | MP5 | 0.89 | 0.76 | 0.52 | 0.22 |
> | DEPS | 0.80 | 0.69  | 0.17 | 0.02 |
> | JARVIS-1 | 0.89 | 0.89 | 0.35 | 0.09 |
> | Optimus-1 | 0.98 | 0.92 | 0.47 | 0.12 |
> | LARM | 1.00 | 0.97 | 0.57 | 0.20 |
>
> ## Q4: Significance from previous works
>
> We believe this work shows great difference and advantages over the works mentioned by the Reviewer.
>
> * The works mentioned by the Reviewer are also based on LLMs. This means if the LLM does not have accurate knowledge about this environment, the method fails. By contrast, ours combines LLM and RL. These two parts both contribute to learning new tasks. If LLM does not have accurate knowledge (the referee reward is noisy), the RL still conducts learning through environment reward. To show this, we have added experiments using a household simulator and a real robot. Refer to the replies to Q1 and Q2 of Reviewer 4ZZV for results.
>
> * The methods mentioned by the Reviewer are all based on a stack of heavy models, meaning slow inference and high deployment cost. However, embodied applications mostly require fast response and deployment on local devices. Our real robot experiment shows the trained policy can be deployed using the local resource of a robot to achieve good success rates.
>
> ## Q5: Unfair comparison
>
> The comparison is absolutely fair. Comparing methods in MineDojo and Mineflayer, our method does not use any extra information than the compared methods. If we call an API, the compared methods also call this API to get information. The inventory list and environment information used by our method are also used in other methods.
>
> ## Q6: How skills are generated
>
> We use a skill generation pipeline similar to Voyager, but the overall method framework is very different. Voyager completely relies on GPT-4, a remote LLM, to complete tasks in an offline way. Ours supports online learning and is lightweight.
>
> ## Q7: Skills in MineDojo
>
> Although the basic actions in MineDojo are low-level actions like moving forward a step, almost all previous works in MineDojo use RL policies and rule-based methods to build higher level skills, like search a tree. The Reviewer can refer to the code of Plan4MC to know how these skills are built. Our skills used in MineDojo completely follow them, therefore the comparison is absolutely fair. We do not use any extra information.
>
> ## Q8: Success Rate 0.65
>
> We test our own method for 30 times to compute the success rate. The 0.65 is the success rate of a compared method and is obtained from its original paper.

---

> > ### Comment · Reviewer_xmFZ · 2025-04-02
> >
> > Thank you for the response. However, I still have the following concern:
> >
> > ## Response to Q1:
> >
> > Although the reviewer understands that this was an oversight, every claim made in the paper should be rigorous and accurate. The reviewer hopes that the authors will revise this claim accordingly in the revision.
> >
> > ## Response to Q2:
> >
> > The reviewer appreciates the authors' efforts in conducting experiments in alternative environments. However, as the main experimental setting of this work, the experiments conducted in Minecraft still lack a sufficient number of evaluation tasks. While the reviewer acknowledges the challenges of conducting experiments on the full MineDojo benchmark, it is worth noting that several prior works have been evaluated on significantly more tasks than LARM, such as 71 tasks for DEPS [NeurIPS'23], 67 tasks for Optimus-1 [NeurIPS'24], and 200 tasks for Jarvis-1 [TPAMI'24].
> >
> > ## Response to Q3:
> >
> > The reviewer appreciates the authors for incorporating the suggested baselines into the revision. However, it is important to note that these baselines should be fairly compared in the main Table 1 of the paper. This raises a concern, as the reviewer observes that LARM underperforms these baselines on many tasks. For example, in the task "harvest stick," LARM achieves a success rate of 0.93, while Jarvis-1 reaches 1.0. For more comparisons, please refer to the original sources mentioned above.
> >
> > ## Response to Q6:
> >
> > The reviewer finds the authors' claim—"Voyager completely relies on GPT-4, a remote LLM, to complete tasks in an offline way. Ours supports online learning and is lightweight."—somewhat unclear. To the best of my knowledge, Voyager generates code (i.e., skills) in an online environment, thereby enabling the continual updating of its skill library. Moreover, the inference phase of Voyager is also conducted in an online setting. Therefore, what is “offline way” for Voyager?
> >
> > ## Response to Q7:
> >
> > Thank you for pointing out Plan4MC, which allowed the reviewer to spend time studying and understanding the implementation details of LARM. However, the reviewer suggests that these details be included in the main text or the appendix, as doing so could help reduce potential confusion and save readers considerable time.
> >
> > The reviewer appreciates the additional experiments and clarifications provided during the rebuttal phase. However, the reviewer does not consider the concerns mentioned above to be "misunderstandings." If these concerns can be adequately addressed, the reviewer is open to reconsidering the score with a positive attitude.

---

> > > ### Author Response · Authors · 2025-04-03
> > >
> > > ## Response to Q1: Rigorous claims
> > >
> > > We greatly thank the Reviewer for all the constructive feedbacks. We will update all the revised claims and experiments to the paper.
> > >
> > > ## Response to Q3: Inferior performance in Table 1
> > >
> > > For more clear explanation, it is better to first address the concern in Response to Q3 (Inferior performance). Afterwards, we address the Response to Q2 (More evaluation tasks).
> > >
> > > The performance of LARM in Table 1 of the paper is inferior to some methods like Javis-1 is because of **unfair comparison**. In fair comparison, LARM outperforms them.
> > >
> > > The research focus of this work and all the works mentioned by Reviewer is high-level skill scheduling. For fair comparison between such two works, we need to make sure two experiment settings are the same, the experiment environment and used low-level skills.
> > >
> > > The experiment environment used by the mentioned works like Javis-1 is the Minecraft Universe Benchmark. By contrast, the experiment environments used in Table 1 and Table 2 of the paper are MineDojo and Mineflayer, respectively. In different experiment environments, the task setting, agent initial status, and agent atom actions can be different. This is why we compare LARM with other methods using two tables in the paper. In Table 1 and Table 2 of our paper, the compared methods in the same table adopt the same environment.
> > >
> > > For low-level skills, the skills used in Table 1 of the paper are based on Plan4MC, while the works mentioned by the Reviewer adopt STEVE-1. The low-level skill policy does not always execute a skill successfully, and the failure of any skill execution could result in the task completion failure of the high-level scheduling policy. The low-level skill policy STEVE-1 adopted by the mentioned works is much stronger than our employed Plan4MC in Table 1 of the paper (the skill execution success rate of STEVE-1 is higher than Plan4MC). Therefore, the success rate reported by Javis-1 is higher than the reported success rate of LARM in Table 1 due to unfair comparison.
> > >
> > > To ensure fair comparison, we re-test LARM using the same experiment setting as the mentioned works (based on Minecraft Universe Benchmark and STEVE-1). The whole experiment results are reported in the Reply to Response to Q2 (the following Reply). In fair comparison, LARM outperforms the mentioned works like Javis-1.
> > >
> > > We will add all these explanations to the paper.
> > >
> > > ## Response to Q2: More evaluation tasks
> > >
> > > As suggested by the Reviewer, we conduct experiments on 200 tasks following Javis-1.
> > >
> > > As explained in the Response to Q3, we test our method  based on the experiment environment Minecraft Universe Benchmark and low-level skill policy STEVE-1, which are consistent with the compared methods. The results are reported in the same format as Javis-1 (group 200 tasks into 7 categories and report the average success rate of each category). The results of the compared methods come from the paper of Optimus-1. The success rates of all these methods are as follows:
> > >
> > > | Method | DEPS | JARVIS-1 | Optimus-1 | LARM |
> > > | :-: | :-: | :-: | :-: | :-: |
> > > Wood | 0.77 | 0.94 | 0.99 | 1.00
> > > Stone | 0.49 | 0.89 | 0.92 | 0.97
> > > Iron | 0.16 | 0.36 | 0.47 | 0.57
> > > Gold | 0.00 | 0.07 | 0.09 | 0.17
> > > Diamond | 0.01 | 0.09 | 0.12 | 0.20
> > > Redstone | 0.00 | 0.16 | 0.25 | 0.30
> > > Armor | 0.10 | 0.16 | 0.19 | 0.27
> > >
> > > According to the results, we can find that LARM outperforms all the compared methods in all categories. This reveals the advantages of combining LLM and RL over solely based on LLM (the compared methods).
> > >
> > > We will add all the descriptions and experiment results to the paper.
> > >
> > > ## Response to Q6: Offline way of Voyager
> > >
> > > We apologize for the unclear explanation. Voyager is offline because its method is not in use after the agent starts to execute scheduled skills.
> > >
> > > Voyager has two phases, exploration and test.
> > >
> > > In exploration, given a target task, it uses GPT-4 to write the code of skills that controls the agent to complete the task. If the task is not executed successfully, GPT-4 is prompted to revise the code until the task is completed successfully.
> > >
> > > In test, Voyager employs GPT-4 to decompose the target task into the order of executing skills generated in exploration. After the task decomposition, the agent executes the code of scheduled skills one by one. **During executing code, Voyager has no perception, reasoning, or a mechanism of revising the code based on new environment observation.** This means if something unexpected in the code happens (this often happens), Voyager cannot handle this problem. This is what "offline way" refers to, its method is not in use after the agent starts to execute scheduled skills.
> > >
> > > By contrast, LARM perceives the environment and reasons about what is the next skill during executing the target task. LARM can adjust its actions in unexpected situations, and thus outperforms Voyager.
> > > ## Response to Q7: Add details
> > >
> > > We will add all these details to the paper as suggested by the Reviewer.

---

### Official Review · Reviewer_2LZg · 2025-03-14

**Overall Recommendation:** 2

**Summary:**

This paper introduces a lightweight LLM-based agent that balances efficiency and generalization for long-horizon tasks. Using Referee RL, which employs a giant LLM for immediate feedback, LARM overcomes reward vanishment in reinforcement learning. Tested in Minecraft, it outperforms previous methods, achieving complex goals like enchanted diamond equipment.

**Claims And Evidence:**

The paper claims that "LARM runs at 0.58 seconds per inference, meeting online inference requirements." However, with an interaction FPS of less than 2, this is insufficient for real-time deployment. The claim should be more accurately framed.

**Essential References Not Discussed:**

This paper does not cite some related works:

[1] Wang et al. Jarvis-1: Open-world Multi-task Agents with Memory-Augmented Multimodal Language Models. T-PAMI 2024.

[2] Jiang et al. CLIP-Guided Reinforcement Learning for Open-Vocabulary Tasks. ECCV 2024.

[3] Li et al. Auto MC-Reward: Automated Dense Reward Design with Large Language Models for Minecraft. CVPR 2024.

[4] Wang et al. OmniJarvis: Unified Vision-Language-Action Tokenization Enables Open-World Instruction Following Agents. NeurIPS 2024.

**Experimental Designs Or Analyses:**

The method is only evaluated on Minecraft, whereas related works typically demonstrate robustness and generalization across multiple gaming environments [4]. Testing on other domains, including games with incomplete Wiki resources (e.g., Montezuma’s Revenge), would strengthen the claims.

When test in MineFlayer, are the skill code are generated by GPT-4 or your fine-tuned models?

The paper lacks evaluation on other vision-language models (e.g., Fuyu, BLIP, Qwen2-VL), making it unclear if LARM's improvements generalize across architectures.

**Methods And Evaluation Criteria:**

The proposed method is highly similar to prior works [2] and [3], which also employ multimodal large models as reward models to assist agent learning. The contributions are incremental rather than groundbreaking.

The paper fails to compare LARM with essential baselines, particularly [2] and [3], which are closely related. A fair comparison is necessary to establish the advantages of the proposed approach.

**Other Comments Or Suggestions:**

See in weakness.

**Other Strengths And Weaknesses:**

The method relies heavily on Minecraft Wiki for learning. However, the paper does not quantify how much this pretraining improves performance over the original model, leaving a significant gap in evaluation. Additionally, the preprocessing details of the Minecraft Wiki corpus are missing, making replication difficult.

**Questions For Authors:**

See in Weakness.

**Relation To Broader Scientific Literature:**

No.

**Theoretical Claims:**

Yes.

---

> ### Author Rebuttal · Authors · 2025-03-31
>
> We have addressed the concerns of the Reviewer one by one in the following. The paper will be revised accordingly.
>
> ## Q1: Real-time inference
>
> The speed of 0.58 second per inference is the inference time of the high-level scheduling policy. Each time of high-level schedule corresponds to seconds of low-level skill execution. The inference speed of the low-level skill policy is more than 1000 FPS. Therefore, we claim that our method meets the online inference requirement.
>
> To avoid misunderstanding, we change the claim in the paper as "meet the speed requirement of online high-level action scheduling".
>
> ## Q2: Similar to two previous works
>
> The two works mentioned by the Reviewer are both about low-level action learning, while ours is about high-level scheduling. The motivation, method, and results are all very different:
>
> * CLIP RL: This work uses CLIP attention map to replace text instruction and provides an invariant representation for the policy to approach different objects. This work does not have significant similarity with ours, which combines LLM and RL.
>
> * Auto MC-Reward: This work relies on LLMs to write code to generate reward functions. Its idea can be applied to learn low-level skills where reward functions can be defined explicitly. However, in many tasks (like the real robot block building task in the Reply to Q2 of the Reviewer 4ZZV), the rewards are too subjective and complex to define explicitly, so this work is inapplicable. By contrast, our method does not have this limitation.
>
> As suggested by the Reviewer, we add comparison to these two works. For fair comparison, all three works adopt the train and test settings of Auto MC-Reward. The success rates on different key achievements are as follows:
>
> | Method | Wood | stone | iron | diamond |
> | :-: | :-: | :-: | :-: | :-: |
> | CLIP RL | 0.64 | 0.23 | 0.02 | 0.00 |
> | Auto MC-Reward | 0.85 | 0.78 | 0.63 | 0.29 |
> | LARM | 0.98 | 0.96 | 0.81 | 0.56 |
>
> ## Q3: Test in more environments
>
> As suggested by the Reviewer, we have test our method in more environments, including VirtualHome (a household activity simulator) and the Cobot Magic robot (a robot in the real world). Due to the reply character limit, please refer to the replies to Q1 and Q2 of Reviewer 4ZZV for experiment details and results. We believe these experiments sufficiently confirm the practical value of our work.
>
> ## Q4: How skills are generated
>
> The skills are generated by GPT-4. Our fine-tuned model is used for online high-level action scheduling. It does not serve as the referee model.
>
> ## Q5: Generalization to more LLMs
>
> Our method can be generalized to different LLMs. As suggested by the Reviewer, we add experiments that test using different LLMs as the referee and LARM base model.
>
> First, we study the success rates adopting different sizes of referee LLMs, including TinyLLaVA-3.1B, Fuyu-8B, Llama3-8B, and Qwen2-14B. Notably, these models do not always provide correct referee reward, meaning the reward quality is different. The success rates are as follows:
>
> | referee | Stick | Wooden | Stone | Iron |
> | :-: | :-: | :-: | :-: | :-: |
> | TinyLLaVA-3.1B | 0.73 | 0.50 | 0.17 | 0.00 |
> | Fuyu-8B | 0.87 | 0.57 | 0.27 | 0.03 |
> | Llama3-8B | 0.90 | 0.67 | 0.33 | 0.13 |
> | Qwen2-14B | 0.90 | 0.67 | 0.37 | 0.20 |
> | GPT-4o | 0.93 | 0.70 | 0.40 | 0.27 |
>
> According to the results, we can observe that generally LLMs with more parameters lead to better performance. The key factor is the quality of the generated referee reward. Refer to the replies to Q1 and Q4 of Reviewer xu7T for more in-depth analysis on the reward noise tolerance of LARM.
>
> Then, we study replace the LARM based model from TinyLLaVA-3.1B
> to Fuyu-8B and Qwen2-14B. The success rates and speed are as follows:
>
> | Base Model | Stick | Wooden | Stone | Iron | Inference Time
> | :-: | :-: | :-: | :-: | :-: | :-: |
> | TinyLLaVA-3.1B | 0.93 | 0.70 | 0.40 | 0.27 | 0.58 |
> | Fuyu-8B | 0.97 | 0.77 | 0.43 | 0.30 | 1.19 |
> | Qwen2-14B | 1.00 | 0.80 | 0.47 | 0.33 | 2.88 |
>
> According to the results, we can observe that replacing the base model with a larger one generally benefits the performance. However, the inference time is also increased significantly.
>
> ## Q6: Missing reference
>
> As suggested, we will add all these references to the paper and discuss the relation with them.
>
> ## Q7: Wiki Pre-train effect
>
> The Wiki pre-train improves model convergence speed and success rates on different tasks significantly. Actually, we have reported the success rate gains caused by Wiki pre-train in Table 4 of the paper. For the convenience of review, we present the results in the following table. Besides success rates, we also report the exploration iteration number for arriving convergence.
>
> | Wiki Pre-train | Exploration Iterations | Stick | Wooden | Stone | Iron |
> | :-: | :-: | :-: | :-: | :-: | :-: |
> | No  | 8500 | 0.83 | 0.57 | 0.33 | 0.13 |
> | Yes | 5000 | 0.93 | 0.70 | 0.40 | 0.27 |
>
> The Wiki data preprocessing pipeline is the same as the work MineDojo.

---

### Official Review · Reviewer_4ZZV · 2025-03-15

**Overall Recommendation:** 3

**Summary:**

This paper focusses on the long-horizon embodied intelligence, specifically, the MineCraft tasks. Previous works generally rely on the strong generalization of giant LLM agents, since the performance of lightweight LLMs such as LLaVA-7B is limited. However, this requires huge computing resources. In this paper, the authors aim to combine the advantages o both RL methods and LLM methods while avoiding their shortcomings. To achieve this, they first propose Large Auto-Regressive Model (LARM), with the main body using the same lightweight LLMs   as TinyLLaVA. LARM is equipped with basic knowledge about the game it is playing by using numerous WiKi webpage data for pre-training. It predicts the next action to perform in an auto-regressive manner by taking environmental observation as input. To train LARM, this paper introduce referee RL instead of using traditional RL which will lead to reward vanishment during long-horizon embodied exploration. The core idea is to employ a referee (like a giant LLM) to provide immediate feedback about whether the just performed action brings positive contribution to realizing the final target.

**Claims And Evidence:**

Yes. this paper provide solid experiments to valid their calim: using a lightweight LARM for long-horizon embodied tasks while using referee RL as the training method.

**Essential References Not Discussed:**

No.

**Experimental Designs Or Analyses:**

Yes. the experiments are extensive.

**Methods And Evaluation Criteria:**

Yes. This paper carries out experiments on Minedojo and Mineflayer, which are commonly used to assess an embodied intelligence's ability of long-term tasks. The author also provides extensive comparisons with previous SoTA methods.

**Other Comments Or Suggestions:**

It will be better if the authors can provide more results in addition to the MineCraft environments. For example, do the authors ever try their methods on simulated household environments like VirtureHome, or even the real world? It will show more value compared to the current bit-style MineCraft, considering the rules of MineCraft are highly structured.

**Other Strengths And Weaknesses:**

This paper use a tiny LLM to finish the long-term diamond task. Comparing with previous works which rely on giant LLMs (e.g., gpt-4o) or finetuned on larger LLMs (e.g., 70B LLaMA), LARM proposed in this paper is time-efficient, which only takes 40 GPU hours on a single RTX4090 GPU.

**Questions For Authors:**

No additional questions

**Relation To Broader Scientific Literature:**

The method proposed in this paper could be potentially adopted to promote AI agent in game environments.

**Theoretical Claims:**

Yes. they are correct.

---

> ### Author Rebuttal · Authors · 2025-03-31
>
> We have test our method in more environments as suggested, and the details are in the following. The paper will be revised accordingly.
>
> ## Q1: Experiment in a household simulator
>
> We thank the Reviewer for this suggestion. As suggested, we conduct experiments in VirtualHome to further validate the effectiveness of our method. We design 10 tasks that each task requires 4 to 6 steps of low-level skill execution. An example of the designed task is "watch TV". To complete this task, the agent needs to complete the following five steps in sequence: (1) Walk to TV. (2) Turn on TV. (3) Walk to a sofa. (4) Sit on the sofa. (5) Watch TV. An anonymous video link describing how LARM performs this "watch TV" task is provided here: [VirtualHome Video](https://drive.google.com/file/d/1Dp0ivtO3e-8xa6fsBtbQD7wLtSl6cYtp/view?usp=sharing).
>
> In this experiment, we remove the Wiki data pre-train step for LARM to show that our algorithm can also work well without Wiki data pre-train. we employ Qwen-VL-Max as the referee model. Notably, Qwen-VL-Max does not always provide correct referee reward (about 10% error rate). Therefore, this experiment shows the robustness of our method based on noisy reward. Refer to the replies to Q1 and Q4 of Reviewer xu7T for more in-depth analysis on the robustness of our method under noisy reward.
>
> In this experiment, we compare our method with classic RL methods including Deep Q-learning, TPRO, and PPO to reveal how much efficiency is improved by combining RL and LLM using our method. The input to policies includes task target description,agent initial state, historical actions. As VirtualHome provides the function to execute each low-level skill, we do not need to implement low-level policies like in Minecraft. For each method, we report the avarage times of explorations the policy needs to complete the designed tasks for the first time (all actions are predicted without random exploration). The results are as follows:
>
> Method | Deep Q-learning | TPRO | PPO | LARM
> | :-: | :-: | :-: | :-: | :-: |
> Average Times | 4876.3 | 3925.8 | 3766.0 | 85.4
>
> The above results sufficiently show the efficiency advantage of LARM. For tasks needing longer action chains (like tens to thousands of steps are needed for the tasks in Minecraft), classic RL methods like Deep Q-learning cannot complete a task for the first time after millions of exploration times. By contrast, LARM learns to complete the task efficiently.
>
> ## Q2: Experiment in the real world
>
> As suggested by the Reviewer, we design an experiment in the real world to further validate the practical value of our method. In this experiment, we train a LARM policy to control a robot to build blocks.
>
> Specifically, several blocks in various shapes are randomly placed on a table. The LARM policy is trained based on our proposed referee RL algorithm to stack blocks as a building similar to a house. To complete this task, the policy needs to learn to select suitable blocks and decide their relative positions to place. The basic skills are implemented based on imitation learning, where we adopt the robotic manipulation algorithm VIRT to grasp and place blocks.
>
> We use Qwen-VL-Max as the referee model. It judges whether the building is becoming more like a house based on image observation. Notably, we do not clearly define how a house should look like. Therefore, there are multiple ways of stacking different blocks to be similar to a house. Correspondingly, the referee reward provided by the referee policy is noisy.
>
> Interestingly, after 100 iterations of random exploration, the policy learns to select and stack different blocks as a house. The anonymous videos of two stacking block process examples are provided here: [Robot Stack Blocks 1](https://drive.google.com/file/d/15dB0SLVZBckz6WzJfIEq7GKv7n_Mugdg/view?usp=sharing) and [Robot Stack Blocks 2](https://drive.google.com/file/d/1SofuQErTk8qn51ChDwSVmTTklY9WpG5X/view?usp=sharing).
>
> According to the results of this experiment, we can conclude that (1) Our proposed method can be applied to a real robot task. (2) By combining LLM and RL, our method can derive policies that are able to perform creative tasks.

---

> > ### Comment · Reviewer_4ZZV · 2025-04-06
> >
> > Thank you for your reply and sorry I make a mistake to use "official comment" instead of "rebuttal comment". I do appreciate the experiments the authors carry out in additional environments like VirtureHome and the reality. For now I hold a positive attitude to this work and will make a final decision after the discussion with other reviewers.

---

### Official Review · Reviewer_xu7T · 2025-03-18

**Overall Recommendation:** 3

**Summary:**

The paper introduces LARM (Large Auto-Regressive Model), a lightweight LLM-based embodied agent designed for long-horizon decision-making in open-world environments.

LARM is built on a lightweight auto-regressive model (fewer than 5B parameters) and directly predicts actions instead of generating text like traditional LLMs, enabling faster inference in real-time settings.
The paper identifies the reward vanishment problem in classic RL, where long-horizon credit assignment becomes ineffective.

To address this, the authors propose Referee RL, a technique where a giant LLM (GPT-4) provides immediate feedback on the quality of actions, distilling generalizable knowledge into LARM without human supervision.

**Claims And Evidence:**

LARM balances efficiency and generalization by combining RL’s efficiency with LLM’s reasoning (Figure 1). It directly predicts actions instead of generating text, enabling faster inference.

LARM outperforms RL and LLM-based baselines (MineAgent, Plan4MC, LLaMA-Rider, RL-GPT), achieving higher success rates in Minecraft tasks and becoming the first AI to craft enchanted diamond equipment (Tables 1 & 2).

Referee RL mitigates reward vanishment, providing immediate feedback via GPT-4, improving long-horizon learning stability (Equation 5, Algorithm 1). Ablations confirm that removing Referee RL hurts performance (Table 3).

**Essential References Not Discussed:**

Decision Transformer (DT) (Chen et al., 2021) is not referenced or discussed.

-DT treats reinforcement learning as sequence modeling, using autoregressive token prediction similar to LARM’s auto-regressive action selection.

-Why it matters: Comparing LARM’s LLM-based policy with DT’s transformer-based decision-making would clarify how Referee RL improves credit assignment compared to DT’s return-conditioned training.

Hierarchical Reinforcement Learning (HRL) methods are not discussed.

-LARM avoids hierarchical task decomposition by relying on an LLM-based policy, but methods like FeUdal Networks (Vezhnevets et al., 2017) or Option-Critic (Bacon et al., 2017) explicitly structure long-horizon tasks with sub-goals.

**Experimental Designs Or Analyses:**

-Strengths:

Comprehensive evaluation of LARM across multiple long-horizon tasks in MineDojo and Mineflayer, demonstrating its effectiveness in open-ended embodied AI.

Comparison with RL-based and LLM-based baselines (MineAgent, Plan4MC, LLaMA-Rider, RL-GPT, Voyager, AutoGPT, STEVE) ensures fair benchmarking (Tables 1 & 2).

-Weaknesses and Missing Analyses:

Referee RL is the key contribution, but its impact is not fully analyzed.

The paper shows that GPT-4-based referee feedback improves LARM, but does not explore alternative reward shaping methods (e.g., inverse RL, reward relabeling).

More experiments are needed to analyze how different referee models (e.g., smaller LLMs vs. GPT-4) affect policy training and success rates.

How sensitive is Referee RL to noisy or incorrect feedback? The paper does not test scenarios where the referee gives suboptimal or misleading rewards.

**Methods And Evaluation Criteria:**

Reasonable methods and evaluation criteria.

LARM uses Referee RL, where GPT-4 provides auxiliary rewards to address reward vanishment in long-horizon tasks.

Built on TinyLLaVA-3.1B with LoRA, it directly predicts actions instead of generating text, enabling faster inference.

Pretrained on a 34GB Wiki dataset for better long-horizon planning.

Evaluation Setup:
Tested in: MineDojo (open-ended AI) and Mineflayer (API-based Minecraft).
Metrics: Task success rates, inference speed (0.58s per step on RTX 4090).

**Other Comments Or Suggestions:**

Referee RL needs further analysis.

How sensitive is Referee RL to incorrect or noisy feedback? The paper does not analyze how GPT-4 errors affect policy learning.
No study on different referee models – The paper assumes GPT-4 is necessary, but smaller LLMs or alternative feedback mechanisms could be explored.

**Other Strengths And Weaknesses:**

-Strengths

Novel approach to handling reward vanishment – LARM introduces Referee RL, where a GPT-4-based referee provides auxiliary rewards, improving long-horizon credit assignment.

Combines RL and LLM advantages while mitigating their drawbacks – Unlike task-specific RL agents or slow, text-generating LLM agents, LARM achieves both efficiency and generalization by directly predicting actions (Figure 1).

-Weaknesses:

The paper is highly specific to VLMs and LLMs, limiting broader RL insights.

The system design and evaluation focus heavily on LLM-driven decision-making, but a more general RL perspective (e.g., comparisons with hierarchical RL, multi-step planning, or alternative reward shaping methods) would provide broader applicability.

Few traditional RL baselines – While the paper compares against LLM-based baselines, it does not compare LARM’s learning efficiency against traditional RL methods beyond PPO.

No discussion on whether Referee RL could benefit non-LLM policies – Could a transformer-based decision model or hierarchical RL agent also benefit from Referee RL?

No computational efficiency analysis.

The paper claims LARM is more efficient than traditional LLM agents, but does not provide FLOP comparisons, memory usage, or training time benchmarks to quantify efficiency gains.

**Questions For Authors:**

How does LARM compare to traditional RL methods beyond PPO?

The paper primarily compares LARM to LLM-based approaches, but how does it compare to hierarchical RL or model-based RL in long-horizon planning?

Impact on Evaluation: If LARM is significantly better than traditional RL, it strengthens the argument for LLM-driven decision-making in embodied AI.
How sensitive is Referee RL to incorrect or noisy feedback?

The paper assumes GPT-4 always provides accurate auxiliary rewards, but what happens if the referee makes errors or inconsistent judgments?

Impact on Evaluation: If Referee RL is highly sensitive to noise, additional robustness measures may be needed.
Could a smaller LLM serve as an effective referee?

The paper uses GPT-4 as the referee, but has the impact of smaller LLMs (e.g., LLaMA-7B, Mistral-7B) been tested?
Impact on Evaluation: If smaller models perform similarly, LARM could be more computationally efficient.

**Relation To Broader Scientific Literature:**

LARM contributes to RL-LLM hybrid learning.

Key novelty:

Referee RL introduces LLM-based auxiliary rewards to mitigate reward vanishment in long-horizon RL.

LARM is an example of multi-modal input models for direct action prediction, showing how LLMs can enhance policy learning without text-based prompting.

What’s missing?

Comparison to hierarchical RL approaches

Discussion on alternative reward shaping methods – hindsight experience replay, etc.

Relation to decision transformers or sequence modeling in RL

**Theoretical Claims:**

LARM identifies the "reward vanishment" problem in long-horizon RL, where TD errors approach zero over time, making standard RL inefficient for credit assignment (Equation 5).

Referee RL mitigates this issue by injecting auxiliary rewards from GPT-4, providing immediate feedback on action quality, which stabilizes long-horizon learning (Algorithm 1, Figure 2).

GAE formulation (Equation 6) supports the claim that standard PPO struggles with long-horizon tasks, while Referee RL preserves meaningful TD errors, enabling more effective optimization.

Discussions:

Missing formal convergence analysis – The paper does not provide a proof that Referee RL leads to stable policy updates over time.

---

> ### Author Rebuttal · Authors · 2025-03-31
>
> We have addressed all concerns of the Reviewer in the following. The paper will be revised accordingly.
>
> ## Q1: Referee RL stable update proof
>
> We thank the Reviewer for this reminder and will add this proof to the paper. Due to the reply character limit, we cannot provide the whole proof here, but we can give a concise description.
>
> PPO can be treated as an extension of TPRO, and the stable update of TPRO has been proved. Thus, to prove the stable update criteria of referee RL, we need to focus on the key difference between referee RL and PPO, the reward noise. To imitate this noise, we replace the correct reward as false reward with a ratio $\sigma$ with respect to a random distribution. Empirically, if $\sigma$ exceeds a threshold $\epsilon$, the policy cannot converge stably. We can analyze the value of $\epsilon$ based on the gradient bias analysis in stochastic optimization theory.
>
> ## Q2: The impact of referee RL and relation with RL techniques
>
> Referee RL can be applied to various RL algorithms beyond PPO.
>
> Referee RL combines the techniques of RL and LLM to address the key problems in these two communities, therefore benefiting both communities.
>
> The key problem of RL is its exploration inefficiency. LLM has general knowledge. Our work shows that LLM can provide guidance to RL and reduce the exploration cost very significantly (like the experiment results in the Reply to Q1 of Reviewer 4ZZV).
>
> The key problem of LLM is it lacks the knowledge of direct environment interaction. Based on RL exploration, we tune a lightweight LLM into a SOTA embodied agent.
>
> Our method does not conflict with the RL techniques mentioned by the Reviewer, such as hierarchical RL. They handle the long-horizon exploration problem from different perspectives and can be applied simultaneously.
>
> ## Q3: Exploration on more RL methods
>
> The RL methods mentioned by the Reviewer include hierarchical RL, inverse RL, reward relabling, Decision Transformer, more RL baselines. These methods do not conflict with our method and can be applied simultaneously. We will add discussion on the relation with these methods to the paper.
>
> Specifically, our LARM model is for high level scheduling, and a part of the low-level skills are implemented based on PPO or deep Q-learning. So hierarchical RL has been used in our work.
>
> Inverse RL assumes there are optimal expert demonstrations. However, the demonstrations are unavailable in our studied task. For reward relabling, it assumes the historical actions are optimal under the reward function. However, this assumption is often too strong to meet, and the reward is also difficult to define explicitly.
>
> We have tried the idea of Decision Transformer, which relies on training on expert demonstration data. To realize this idea, we first collect abundant demonstration data, and the method achieves similar performance as referee RL with less computing cost (4 hours of training). However, the expert demonstration data is often unavailable.
>
> We have tried replacing the PPO in referee RL as deep Q-learning, and the result comparison is as follows:
>
> |  | Stick | Wooden | Stone | Iron |
> | :-: | :-: | :-: | :-: | :-: |
> | Deep Q-learning | 0.87 | 0.63 | 0.30 | 0.17 |
> | PPO | 0.93 | 0.70 | 0.40 | 0.27 |
>
> We can see that PPO based referee RL gets better performance.
>
> ## Q4: Influence of referee reward noise
>
> We have analyzed reward noise influence theoretically in the reply to Q1. This part analyzes it using experiments. We randomly modify the reward generated by GPT-4 with a ratio $\sigma$, and the success rates on different tasks are as follows:
>
> | $\sigma$ | Stick | Wooden | Stone | Iron |
> | :-: | :-: | :-: | :-: | :-: |
> | 0% | 0.93 | 0.70 | 0.40 | 0.27 |
> | 10% | 0.93 | 0.67 | 0.30 | 0.17 |
> | 30% | 0.77 | 0.33 | 0.17 | 0.07 |
> | 50% | 0.50 | 0.13 | 0.00 | 0.00 |
>
> We can observe that (1) Stronger noise deteriortes the sucess rates. (2) Tasks requiring longer execution chains are less noise tolerant.
>
> In addition, we design two new experiments using the VirtualHome simulator and a real robot, where the referee LLM sometimes provide incorrect rewards. Refer to the replies to Q1 and Q2 of Reviewer 4ZZV for experiment details and results.
>
> ## Q5: Results on smaller referee LLMs
>
> As suggested by the Reviewer, we study the success rates adopting different sizes of referee LLMs. Due to the reply character limit, refer to the replies to Q5 of Reviewer 2LZg for details.
>
> ## Q6: Computing efficiency analysis
>
> The reason we did not provide quatitative efficiency comparison with previous LLM based methods is these methods are often based on large LLMs deployed on remote servers, like GPT-4. We cannot know their specific parameter numbers, FLOPs, etc. What we can make sure is their computing costs are enormous as publicly known and far more than ours. We report the efficiency metrics of our method as follows:
>
> Inference Time | Inference Memory | FLOPs | Training Time
> | :-: | :-: | :-: | :-: |
> 0.58s | 3.8G | 614.8G | 42 hours

---

### Decision · Program_Chairs · 2025-05-01

**Decision:**

Accept (poster)

**Comment:**

This paper proposes the Large Auto-Regressive Model (LARM), an autoregressive LLM-based agent for open-world decision-making problems such as Minecraft. The main aspect of the LLM training is Referee RL, which uses a large language model (GPT-4 class) to provide feedback on action quality. Results are demonstrated on Minedojo and Mineflayer.

  The paper received borderline scores, with appreciation for the interesting method and analysis as well as thorough comparison. Some weaknesses were mentioned, however, for example lack of in-depth analysis/ablation of the components (e.g. Referee RL), comparison to other reward shaping and RL methods, analysis on suboptimal feedback, and additional experiments. The authors provided a comprehensive rebuttal, including additional comparisons (CLIP RL, MC-Reward, etc.) and new experiments in VirtualHome and real-world experiments. After significant back and forth, the review scores remain borderline.

  Having considered the paper, reviews, rebuttal, and significant discussion, I lean towards acceptance of this paper. The method provides an interesting analysis and method with RL-based policy learning, and the rebuttals address most of the concerns. Therefore, I believe that the positives outweigh the remaining weaknesses, and this paper would be an interesting contribution. I strongly encourage the authors to include the results of the discussion, including new experiments. Further, I find that the paper overall lacks a situation with respect to vision-language-action models, which also finetune (often larger) VLMs and besides supervised fine-tuning have begun to include RL-based finetuning as well to yield generalist embodied agents.